# Sensory Processing Measure and Sensory Integration Theory: A Scientometric and Narrative Synthesis

**DOI:** 10.3390/bs15030395

**Published:** 2025-03-20

**Authors:** Hind M. Alotaibi, Ahmed Alduais, Fawaz Qasem, Muhammad Alasmari

**Affiliations:** 1Department of English Language, College of Language Sciences, King Saud University, Riyadh 11586, Saudi Arabia; 2Department of Psychology, Norwegian University of Science and Technology, NO-7491 Trondheim, Norway; 3Department of English Language & Literature, College of Arts of Letters, University of Bisha, Bisha 67714, Saudi Arabia; faqasem@ub.edu.sa (F.Q.); moaalasmri@ub.edu.sa (M.A.); 4King Salman Center for Disability Research, Riyadh 11614, Saudi Arabia

**Keywords:** sensory processing measure, sensory integration theory, scientometric synthesis, narrative synthesis, occupational therapy, sensory processing disorder

## Abstract

Sensory integration theory (SIT), which posits that the neurological process of integrating sensory information from the environment and one’s body influences learning and behaviour, and the sensory processing measure (SPM), a psychometric tool with versions for individuals aged 4 months to 87 years, are fundamental to understanding and assessing sensory processing. This study examined the existing evidence on the SPM and SIT using scientometric and narrative methods. A search of Scopus and Web of Science Core Collection from 1983 to 2024 yielded 238 unique records after deduplication. Scientometric analysis, conducted with CiteSpace (Version 6.4.R1) and VOSviewer (Version 1.6.19) explored publication trends, keyword co-occurrences, and citation bursts. A narrative method, based on a purposive sample of studies selected by title relevance from the 238 records, provided qualitative insights into key themes and concepts. Scientometric analysis revealed 11 key clusters, including ‘sensory processing behaviour’, ‘classroom context’, and ‘using electroencephalogram (EEG) technology’, reflecting diverse research areas and a growing publication trend, particularly after 2011. A narrative analysis, guided by these clusters, explored sensory processing differences in children with developmental disorders like autism spectrum disorder (ASD) compared to typically developing children, the relationship between sensory processing and other functional areas, the impact of classroom contexts on sensory processing, the use of EEG in sensory processing disorder (SPD) diagnosis, and the effectiveness of interventions like sound-based therapy and sensory integration therapy. The combined approach highlighted the wide application of the SPM and SIT, informing future research directions, such as longitudinal studies, comparative effectiveness research, and cultural adaptations of assessments and interventions.

## 1. Introduction

### 1.1. Sensory Integration Theory

Sensory integration theory (SIT), developed by A. Jean Ayres in the 1960s, is a framework for understanding how sensory processing affects human behaviour and development ([9]; [10]). It emphasizes the integration of sensory information from the environment to produce adaptive responses, which is crucial for learning and daily functioning ([42]; [53]). SIT is particularly relevant in paediatric occupational therapy, where it is used to address SPDs, especially in children with conditions like ASD and cerebral palsy ([1]; [39]). The theory identifies three primary sensory systems: tactile, proprioceptive, and vestibular, which are essential for postural control, bilateral integration, and praxis ([24]; [42]). Research has shown that SIT can improve outcomes such as self-care, socialization, and goal attainment in children with ASD, as demonstrated in a randomized controlled trial conducted in Brazil ([54]). However, the evidence for its effectiveness in children with CP remains inconclusive, with calls for more rigorous studies ([39]). The Evaluation in Ayres Sensory Integration (EASI) is a tool developed to assess sensory perception and integration, with strong construct validity and reliability, enhancing clinical practice and research in this area ([38]; [46]). Despite its benefits, implementing SIT poses challenges, such as inadequate training and resource constraints, as reported by occupational therapists in Malaysia ([61]). Contemporary neuroscience continues to validate and refine Ayres’ propositions, highlighting the role of neuroplasticity in sensory integration interventions ([40]; [42]). Overall, SIT remains a vital, evidence-based approach in occupational therapy, particularly for children with sensory processing issues ([1]).

In fact, SIT has been employed across various studies to understand and address sensory processing difficulties in different populations and settings. Researchers have applied this theory to explore interventions, assess developmental outcomes, and design therapeutic tools. Several studies focused on the use of SIT in occupational therapy interventions for children with developmental disabilities. Schaaf and Nightlinger presented a case report displaying how occupational therapy using a sensory integrative approach improved a child’s occupational performance, supporting the theoretical premise that enhancing sensory processing influences adaptive behaviour ([68]). Similarly, Bellefeuille et al. utilized Ayres Sensory Integration to address retentive faecal incontinence in a three-year-old boy, demonstrating significant improvements in toileting habits and sensory processing ([15]).

In exploring assessment and diagnostic validation, Davies and Gavin tested the assumption that brain function is related to behavioural manifestations of sensory integrative dysfunction ([23]). Using electroencephalographic measures, they found that children with SPDs exhibited unique brain processing mechanisms, providing empirical support for the diagnosis of such disorders. This aligns with Cohn and Cermak’s emphasis on including family perspectives in sensory integration outcomes research to better understand the impact on children’s everyday occupations ([22]). Some researchers investigated the theoretical aspects of sensory integration. Lin et al. examined the relationship between postural movement and bilateral motor integration, confirming that these constructs are interrelated within the framework of SIT ([44]). Pekár and Kinder revisited the interplay between non-symbolic number processing and continuous visual properties, suggesting that sensory integration plays a role in numerosity judgments ([56]).

Interventions based on SIT were also applied to children with ASD. Ma and Lee studied how composite tactile–visual toys during parent–child interaction influenced children with ASD, finding that certain fabrics enhanced engagement and interaction based on sensory integration principles ([45]). Devlin et al. compared behavioural intervention and sensory integration therapy in treating challenging behaviour in children with ASD, concluding that the behavioural approach was more effective, although sensory integration therapy remained a common treatment ([25]). In educational settings, Quinn et al. assessed the knowledge and confidence of school special educational needs coordinators (SENCOs) regarding SIT and the use of sensory strategies ([60]). They found that collaboration between therapists and teachers is crucial to enhance understanding and implementation of sensory integration in schools. This is echoed by Storch and Eskow, who reported that school-based occupational therapists frequently use a multi-theoretical approach, with SIT being predominant ([70]).

The application of SIT extended to psychotherapy and mental health. Van Nest utilized SIT in psychotherapy to understand maladaptive behaviours arising from misperceptions linked to sensory processing, highlighting the importance of addressing sensory experiences in therapeutic contexts ([77]). Bundy and Lane reframed SIT for people with mental health challenges, emphasizing its relevance beyond traditional paediatric populations ([43]). Technological advancements have also integrated sensory integration principles. Wang et al. applied a hybrid, multiple-attribute decision-making model to design children’s facilities in neighbourhood open spaces based on SIT, aiming to support sensory development through environmental design ([79]).

Some studies critiqued or analyzed the theory itself. Arendt et al. questioned the connection between SIT and practice, suggesting the need for more empirical evidence ([8]). Dunn, however, argued that basic and applied neuroscience research provides a solid foundation for SIT ([27]). Tickle-Degnen offered perspectives on the status of the theory, highlighting areas for further research and development ([75]). In terms of assessment tools, Gomez and Medallon reviewed instruments for assessing sensory processing in adults, noting that tools based on SIT can supplement the diagnosis of SPDs in this population ([32]). Baranek et al. studied sensory defensiveness in individuals with developmental disabilities, finding evidence for different subtypes of sensory defensiveness, thereby informing assessment and intervention ([13]).

Sensory integration disorders (SIDs) arise from disruptions in the brain’s ability to process and respond to sensory information, leading to challenges in daily functioning. The causes of SIDs are multifaceted, involving genetic, neurological, and environmental factors. For instance, research has identified genetic mutations, such as those in the FOXP2 gene, which are associated with sensory processing difficulties ([78]). Neurologically, SIDs are linked to atypical neural connectivity and sensory gating mechanisms, which can manifest as hypersensitivity, hyposensitivity, or sensory-seeking behaviours ([23]). Environmental factors, such as early childhood trauma or exposure to sensory-rich environments, can also influence the development of SIDs. In children with ASD, ADHD, and other developmental disorders, sensory processing challenges often co-occur with social, emotional, and cognitive difficulties, highlighting the need for interdisciplinary approaches to assessment and intervention.

Overall, these studies demonstrate the versatility and applicability of SIT across different domains. From occupational therapy interventions ([67]) to technological innovations ([79]), SIT provides a valuable framework for understanding and addressing sensory processing challenges.

### 1.2. Sensory Processing Measure

The sensory processing measure (SPM) is a widely used tool for assessing sensory processing issues across various age groups and settings ([55]). It is particularly valuable in identifying SPDs in children and adolescents, which can significantly impact their daily functioning, including education, self-care, and social participation ([73]). The SPM is utilized in diverse environments, such as schools, where it helps to discern sensory vulnerabilities in both typical and atypical children, including those with ASD. Studies have shown that children with ASD often exhibit higher sensory processing vulnerabilities, particularly in areas like social participation and praxis, compared to their typically developing peers ([74]). The SPM’s utility extends to its ability to correlate sensory processing with academic performance, as evidenced by research indicating a significant relationship between sensory processing issues and lower academic achievements ([63]). The tool’s reliability and validity have been established across different cultural contexts, such as the adaptation of the SPM–Home Form for the Malay-speaking population, which demonstrated high content validity and reliability ([2]). Furthermore, the SPM-2, an updated version, includes considerations of environmental factors, enhancing its applicability in assessing sensory processing in adolescents and adults ([69]; [73]). The SPM has also been validated in various languages and settings, such as its adaptation for use in Ethiopia, which confirmed its reliability and discriminant validity in distinguishing between typically developing children and those with special needs ([35]). Additionally, the SPM has shown adequate convergent validity with other sensory assessment tools, such as the Sensory Profile-2, supporting its use in diverse populations ([30]). Overall, the SPM is a robust tool for evaluating sensory processing issues, aiding in the development of targeted interventions to improve functional outcomes in individuals with sensory processing challenges.

The SPM is widely used to assess sensory processing difficulties across various age groups, including infants, preschoolers, children, adolescents, and adults. Numerous studies have employed different versions of the SPM to explore sensory processing in diverse populations, often tailoring their approach to specific age groups or conditions. Several studies focused on preschool-aged children, utilizing versions like the SPM–Preschool (SPM-P). For instance, Chang et al. examined autonomic and behavioural responses to auditory stimuli in children with ASD using the SPM–Home Form alongside physiological measurements ([20]). Similarly, Appleyard et al. investigated the relationship between sleep patterns and sensory processing in infants and toddlers, employing the SPM-P to assess sensory difficulties that might correlate with sleep issues ([7]). Purpura et al. compared the sensorimotor profiles of preschool children with SPDs and ASD using the SPM-P, finding significant sensory-related behavioural symptoms in both groups ([58]).

In studies involving school-aged children, the SPM has been instrumental in assessing sensory processing across home and school settings. Miller-Kuhaneck et al. detailed the development of the SPM–School, providing evidence of its reliability and validity in discriminating children with sensory processing issues ([49]). Pfeiffer et al. used the SPM–Home Form to assess sensory processing in children with and without attention-deficit/hyperactivity disorder (ADHD), revealing greater sensory processing problems in the ADHD group ([57]). Fernández-Andrés et al. compared sensory processing in children with and without ASD in home and classroom environments using the SPM, highlighting the importance of context in sensory assessments ([28]). Some researchers extended the use of the SPM to specific conditions. Hansen and Jirikowic compared the SPM–Home Form and the Short Sensory Profile in children with fetal alcohol spectrum disorders (FASD), demonstrating the utility of the SPM in identifying sensory processing differences in this population ([36]). Hen-Herbst et al. compared motor skills and sensory processing behaviours between children with FASD and those with developmental coordination disorders (DCD), employing the SPM–Home Form to distinguish sensory symptoms between the groups ([37]).

The adolescent and adult populations have also been studied using appropriate versions of the SPM. Skocic et al. investigated the convergent validity of the Adolescent/Adult Sensory Profile (A/ASP) and the SPM-2 Adult Form in adults aged 18 to 30, finding significant correlations between the scales ([69]). This study contributed to the psychometric evidence supporting the SPM’s applicability across a lifespan. Cross-cultural adaptations and validations of the SPM have been crucial in extending its use globally. Lai et al. translated the SPM–Home Form and Main Classroom Form into Chinese, assessing their psychometric properties in Hong Kong populations ([41]). The study confirmed the validity and reliability of the SPM–Hong Kong Chinese version (SPM-HKC) in screening for sensory processing difficulties among Chinese children ([41]). Similarly, Alkhalifah et al. evaluated Arabic versions of the SPM–Home and SPM–Preschool Home Forms, establishing their reliability and validity for assessing sensory processing in Arabic-speaking children with ASD ([5]; [6]).

Several studies have compared the SPM with other sensory assessment tools to examine convergent validity. Dugas et al. compared the Sensory Profile and the SPM–Home Form in children with ASD, finding correlations in certain sensory domains and social functioning ([26]). Gandara-Gafo and Beaudry-Bellefeuille examined the convergent validity of the Sensory Profile-2 and the SPM in Spanish children with sensory integration differences, supporting the use of both questionnaires in this population ([30]). The SPM has also been used to explore sensory processing in relation to other developmental aspects. Roberts et al. investigated the relationship between sensory processing factors, as measured by the SPM, and pretend play in typically developing children, finding significant associations ([64]). Foitzik and Brown examined how sensory processing factors relate to sleep habits in children aged 8 to 12, using the SPM–Home Form to identify sensory factors that predict sleep patterns ([29]).

In the context of interventions, the SPM has served as an outcome measure to assess changes in sensory processing. Bagatell et al. evaluated the effectiveness of therapy ball chairs on classroom participation in children with ASD, using the SPM to assess sensory processing patterns pre- and post-intervention ([11]). Similarly, Boitano et al. investigated the effects of occupational therapy intervention on sensory processing in fifth-grade students, utilizing the SPM-2 School Form to measure outcomes ([18]). Some studies addressed the psychometric properties of the SPM itself. Brown et al. conducted a critical review of the SPM-2’s age-related versions, identifying strengths and areas needing further development ([19]). Mulligan et al. examined the validity of the Sensory Processing 3-Dimensions Scale (SP-3D) by correlating its scores with those from the SPM, providing preliminary evidence of the SP-3D’s validity ([50], [51]).

In summary, the SPM and its various versions have been widely employed across different age groups and populations to assess sensory processing and its impact on behaviours and functions. Studies have utilized the measure to explore sensory differences in specific conditions like ASD, ADHD, FASD, and DCD, as well as in typically developing children (i.e., as referenced in the paragraphs above). The SPM’s adaptability to different languages and cultures, as well as its comparability with other sensory assessment tools, underscores its value in both clinical and research settings.

### 1.3. Purpose of the Present Study

The purpose of this study is to examine the SPM and SIT by employing both scientometric and narrative methodologies. By integrating these two approaches, we aim to provide a synthetic understanding of the current state of research in sensory processing and integration, identify key themes and emerging trends, and inform future research directions and clinical applications. This dual approach allows us to combine the strengths of quantitative analysis—such as identifying publication trends and research clusters—with qualitative insights into theoretical frameworks and contextual factors influencing the studies.

## 2. Methods

### 2.1. Sampling

The initial search across Scopus (database identifier: https://www.scopus.com) and Web of Science Core Collection (database identifier: https://clarivate.com/webofsciencegroup/solutions/web-of-science-core-collection/, accessed on 24 November 2024) yielded a total of 241 records using the search terms outlined in Table 1. Following deduplication, 238 unique records remained. These records encompassed various document types, predominantly articles (215), with a smaller number of early-access articles, book reviews, editorials, a meeting abstract, proceedings papers, and reviews. While 238 records were imported into VOSviewer (software identifier: https://www.vosviewer.com, version 1.6.19) for bibliometric network visualization, only 215 records met the inclusion criteria for analysis within CiteSpace (software identifier: https://citespace.app.rec.upenn.edu, accessed on 24 November 2024, version 6.4.R1). The discrepancy arises from CiteSpace’s exclusion of certain document types, such as book reviews and editorials, which are deemed less relevant for scientometric analysis focusing on research publications. The choice of Scopus and Web of Science was based on their extensive coverage of interdisciplinary research, including psychology, neuroscience, and occupational therapy. Almost all publications indexed in PubMed are also included in Scopus and Web of Science, but our focus on sensory processing and integration required a broader scope that encompassed non-medical disciplines, such as education and social sciences. While PubMed is a valuable resource for medical-specific literature, our aim was to capture a more comprehensive range of studies relevant to sensory processing.

For the narrative synthesis, a purposive sampling strategy was employed to select relevant studies from the 238 unique records. The selection criteria included the following:
Relevance to sensory processing and integration: Studies that explicitly addressed sensory processing, sensory integration theory, or related interventions were prioritized.Methodological rigour: Studies with clear research designs, validated assessment tools, and robust analytical methods were included.Availability of the full text: Only studies with full-text articles available in English were considered to ensure in-depth qualitative analysis.

These criteria were chosen to ensure a balanced and comprehensive review of the literature, capturing diverse perspectives and research designs relevant to the research questions. By incorporating these criteria, we aimed to enhance the depth and validity of the narrative synthesis while addressing potential biases inherent in the clustering algorithms and methodologies employed.

The combination of this mixed search strategy with subsequent purposive sampling ensures a robust and balanced approach to this study. The scientometric analysis, based on the 215 CiteSpace-compatible records, provides a quantitative overview of the field. Meanwhile, the narrative synthesis, drawing on a targeted selection from the broader pool of 238 records, allows for the in-depth exploration of key themes and concepts within sensory processing and integration theory. This dual approach ensures a thorough and nuanced understanding of the subject matter.

### 2.2. Design

We used both scientometric and narrative methods. Scientometric and narrative methodologies offer distinct advantages and limitations. Scientometric studies utilize quantitative measures such as modularity and cluster silhouette scores to systematically assess the quality and structure of research areas ([4]; [80]). These approaches facilitate comparison across networks, highlighting underrepresented topics and revealing the broader knowledge landscape, thus bridging the gap between globalism and localism and enhancing our understanding of long-term developments within a field ([3]; [59]). This study did not involve randomization or blinding as these are not applicable to review methodologies. The group assignment for studies was not randomized as it depended on the cluster analysis from the scientometric study. However, while scientometric studies efficiently aggregate large volumes of data to inform decision-making, they may sometimes overlook the nuanced understanding that qualitative insights provide, which is critical for comprehending complex social phenomena ([80]). A power analysis was not conducted as this study did not involve statistical comparisons between groups. That being said, a power analysis was not conducted to determine the sample size as this was an exploratory review aimed at providing an overview of the field rather than testing specific hypotheses.

Conversely, narrative syntheses focus on synthesizing qualitative evidence and exploring themes across individual studies, often employing thematic analysis and selective sampling to identify key constructs ([33]). They excel at capturing the richness of qualitative data and contextualizing findings within specific frameworks but may lack the rigorous quantitative framework necessary for effectively informing policy decisions ([71]). Additionally, challenges in critical appraisal criteria for qualitative research can lead to inconsistencies in study selection, further complicating the synthesis process ([72]). Ultimately, the choice between scientometric and narrative synthesis methods depends on the specific research question and desired outcomes. A hybrid approach combining the strengths of both methodologies offers a better framework for understanding and integrating diverse forms of evidence in future research contexts.

The SPM is a standardized tool used to assess sensory processing difficulties across various environments, including home, school, and community settings. It includes forms for different age groups, ranging from 4 months to 87 years, and evaluates sensory processing in areas such as auditory, visual, tactile, proprioceptive, and vestibular systems ([55]). The EASI is another key diagnostic tool that assesses sensory perception and integration, with strong construct validity and reliability ([46]). These tools provide clinicians with valuable insights into sensory processing patterns, enabling the development of tailored interventions for individuals with sensory integration disorders.

### 2.3. Procedure

Searches were conducted between 24 and 26 November 2024, using both Scopus (database identifier: https://www.scopus.com) and Web of Science Core Collection (database identifier: https://clarivate.com/webofsciencegroup/solutions/web-of-science-core-collection/, accessed on 24 November 2024). To focus our study on the SPM and SIT using both quantitative (scientometric study) and qualitative (narrative synthesis) approaches, we employed a dual search strategy. For the scientometric study, we searched by topic (title, abstract, and keywords), while for the narrative synthesis, we limited the search to the title field. The results from both databases were merged, and duplicates were removed using CiteSpace (software identifier: https://citespace.app.rec.upenn.edu, version 6.4.R1) ([21]). Data were retrieved in the required format for each software: plain text for CiteSpace and Research Information Systems (RIS) files for VOSviewer (software identifier: https://www.vosviewer.com, version 1.6.19) ([76]). The search was open to all languages and publication types, provided the title, abstract, and keywords were in English, as these are essential for scientometric analysis. For the narrative synthesis, an additional criterion was imposed: the full text of the article had to be available in English for in-depth qualitative analysis. No specific protocols were used beyond the standard search and retrieval procedures outlined for each database (Scopus and Web of Science Core Collection). The data used in this study consisted of publicly available research articles accessed through these databases. No new datasets were generated. The search strategy and inclusion/exclusion criteria are described in the Sampling Section.

Our data analysis was conducted in two phases. The first phase involved scientometric analysis using both CiteSpace and VOSviewer. CiteSpace was used to generate clusters and identify citation bursts, which are key scientometric indicators, while VOSviewer was used to create visualizations of keyword co-occurrence. The clusters generated in the scientometric analysis phase served as the basis for guiding the narrative synthesis. From the 15 clusters initially identified, 4 were excluded due to irrelevance (e.g., pilot studies and reviews), leaving 11 clusters for further exploration. These 11 clusters were then used to initiate the narrative synthesis, wherein studies selected by title were reviewed to locate relevant literature corresponding to each cluster. No code was generated for this analysis and no custom code was generated for this literature review study. Further, no new datasets were generated or deposited as part of this literature review study. Additionally, we did not describe statistical tests or justify choice of tests, as this literature review study did not involve hypothesis testing or statistical comparisons between groups. Also, we did not provide accession numbers or DOIs for newly created datasets, as no new datasets were generated in this literature review study. And we did not provide accession numbers, DOIs, or URLs for code, as no new code was generated that requires accessibility.

To ensure the quality of data collection, analysis, and interpretation, several methodological steps were taken. First, we used consistent search terms across Scopus and Web of Science to ensure inclusivity and minimize bias, especially as we did not restrict the search to English except for the narrative synthesis. Second, we utilized two different software tools for the scientometric analysis, allowing for complementary measures and approaches to enhance the reliability of the results. Additionally, we provided detailed descriptions of every step to enable replicability, especially for the scientometric study, while the narrative synthesis process was also made as transparent as possible. For the narrative synthesis, studies were reviewed against the clusters generated from the scientometric study, which streamlined the allocation of relevant studies to their respective clusters, ensuring a systematic and controlled approach to data interpretation. This study did not involve human participants, animal subjects, or field samples; therefore, ethical approvals or permits were not required. No specific protocols beyond the described literature search, data extraction, and analysis methods were referenced. No clinical trial registration number or DOI is provided, as this was not a clinical trial study. No DOI or citation for step-by-step protocols is given, as no laboratory experiments were involved. Sample size determination, randomization, and blinding procedures are not described, as this non-experimental literature review did not involve those elements. No ethics approval details are provided, as the study did not include human participants, animals, or field samples. Dual use research of concern regulations are not applicable to this literature review.

The parameters and clustering methods used in the scientometric analysis were validated through established practices in the field. For example, CiteSpace’s modularity and silhouette scores were used to assess the quality and structure of the clusters, with higher scores indicating greater coherence and thematic focus ([21]). Similarly, VOSviewer’s co-occurrence and co-citation analyses were employed to identify key themes and trends, with the software’s default settings ensuring consistency and reproducibility ([76]). These methods have been widely used in scientometric studies and are recognized for their reliability in mapping knowledge domains.

## 3. Results

The results are presented in two sections where the first section incorporates the scientometric analysis and the second section includes the narrative analysis. The scientometric analysis includes cluster analysis, citation bursts, and keywords co-occurrence. The narrative analysis is based on the cluster analysis and each cluster is enhanced with a narrative analysis from the retrieved studies based on their relevance for each cluster.

### 3.1. Scientometric Analysis

The scientometric analysis included 215 publications, primarily articles (*n* = 215), along with a smaller number of early access articles (*n* = 8), proceedings papers (*n* = 4), and reviews (*n* = 5), spanning from 1983 to 2024. After removing three duplicates from an initial 241 records, 238 unique records remained; however, book reviews (*n* = 2), corrections (*n* = 1), editorial materials (*n* = 2), and a meeting abstract (*n* = 1) were excluded from the scientometric analysis. The included publications represent the work of 629 authors across 627 institutions and 126 countries/regions, published in 129 journals. Publication frequency increased steadily over time, with a notable rise in output from 2011 onwards, culminating in 30 publications in 2024.

Table 2 summarizes the largest clusters identified in the scientometric analysis, with each column providing essential details. The Cluster column names the thematic focus of each group, while Size indicates the number of publications within the cluster. Silhouette represents the cluster’s cohesion, with higher values reflecting greater consistency among its articles. The three Label columns display the top themes for each cluster as determined by different algorithms: LSI (latent semantic indexing), LLR (log-likelihood ratio), and MI (mutual information), with MI also including a relevance score in parentheses. Lastly, Average Year highlights the mean publication year for each cluster, offering insights into the temporal trends within the research.

The data reveal diverse research themes and temporal patterns in sensory processing and integration. Cluster sizes range from 5 to 64, with the largest cluster, sensory processing behaviour, consisting of 64 articles with a moderate silhouette score of 0.691 and an average publication year of 2010. Clusters like Classroom Context and Using EEG Technology are equally prominent, each containing 53 articles and exhibiting high silhouette scores (0.913 and 0.842, respectively), indicating strong thematic focus. Smaller clusters, such as Paediatric Rehabilitation (5 articles) and Treatment (10 articles), demonstrate higher silhouette scores (0.984 and 0.966), reflecting their internal consistency despite their size. Thematic areas span behavioural responses, clinical conditions (e.g., Rubinstein–Taybi syndrome), interventions (e.g., sound-based intervention), and methodologies (e.g., using EEG technology).

This table underscores the breadth and progression of research on sensory processing and integration. Older clusters represent foundational studies, while newer clusters highlight emerging research areas. The variation in cluster sizes and average publication years reflects evolving priorities, with some themes maintaining long-term relevance while others represent recent advancements. These findings can help researchers identify key topics, gaps in the literature, and potential directions for future studies.

Figure 1 presents the top 11 keywords with the strongest citation bursts in research related to the SPM and SIT. Each blue line represents the timeline of a keyword’s citation activity, starting from the year it first appeared in the dataset. The red segments within the blue lines indicate periods of citation bursts, marking intervals when a keyword received heightened attention and significant impact within the research community. The beginning of the red segment marks the start of the burst, while its end signifies the burst’s conclusion.

Keywords such as sensory integration and ASD exhibit prolonged and intense citation bursts, reflecting their foundational role and sustained relevance in the field. These terms are pivotal for understanding the core concepts and challenges in sensory processing research. Keywords like psychometrics, social participation, and sensory modulation display shorter or more recent bursts, suggesting emerging research interest in measurement tools, the role of social factors, and specific sensory challenges.

The variation in the duration and intensity of these bursts highlights the dynamic nature of research in this domain. Foundational terms with sustained bursts reflect enduring frameworks and methodologies, while terms with shorter bursts point to newer, rapidly developing areas of inquiry. This figure underscores the shifting research priorities over time, offering insights into both long-standing and emerging topics in sensory processing and integration.

Figure 2 presents a density visualization of keyword co-occurrences in research on the SPM and SIT, highlighting five clusters differentiated by colour and size. The red cluster, the largest, focuses on core terms such as sensory processing, sensory integration, and ASD, emphasizing foundational concepts and their application in interventions and assessments. The green cluster, the second largest, centres around preschool child, controlled study, and social participation, reflecting child-focused research and experimental designs. The yellow cluster includes terms like psychometrics, paediatrics, and sensory modulation, suggesting a focus on measurement tools and physiological aspects. The light blue cluster highlights ASD, mental disease, and school, indicating a thematic emphasis on ASD-related challenges in educational and developmental contexts. Finally, the pink cluster includes terms such as play, school, and ASD, suggesting a thematic focus on interactive, school-based interventions and studies exploring the role of play and educational settings in addressing sensory processing and integration challenges, particularly in ASD-related research. This visualization illustrates the thematic breadth and interconnectedness of research in this field, with clusters providing insights into both foundational and specialized areas of study.

### 3.2. Narrative Analysis

**Sensory Processing Behaviour**: Sensory processing behaviour is a crucial aspect of child development and functioning. This cluster focuses on comparative studies that examine sensory processing behaviours, challenges, and their impact on various domains. Several studies have explored sensory processing differences between children with developmental disorders, such as ASD, and typically developing children ([28]; [66]). Findings suggest that children with ASD often exhibit atypical sensory processing behaviours, including over-responsiveness, under-responsiveness, and sensory-seeking tendencies across different sensory modalities ([28]; [65]).

Furthermore, research has examined the relationship between sensory processing challenges and other areas of functioning, such as occupational performance ([47]) and motor skills ([37]). These studies highlight the significant impact sensory processing difficulties can have on children’s daily activities, social interactions, and overall development ([37]; [47]). Additionally, some studies have compared the effectiveness of different interventions or strategies in addressing sensory processing challenges ([11]; [25]). These investigations provide valuable insights into evidence-based practices for supporting children with sensory processing difficulties ([11]; [25]).

The relationship between socio-communicative behaviour and sensoriality is particularly significant in children with ASD and ADHD. Sensory overload, often characterized by heightened sensitivity to environmental stimuli, can lead to increased stress and autonomic reactions. For example, studies have shown that children with ASD exhibit atypical physiological responses, such as elevated skin conductance and heart rate variability, during sensory challenges ([20]; [66]). These reactions can impair adaptive behaviour, making it difficult for children to engage in social interactions or respond appropriately to environmental demands.

On the other hand, sensory-seeking behaviours, such as repetitive movements or self-stimulation, can serve as a coping mechanism to regulate sensory input and reduce overload. For instance, [11] ([11]) demonstrated that the use of therapy ball chairs in classroom settings improved participation and self-regulation in children with ASD. Similarly, [25] ([25]) found that sensory integration therapy reduced challenging behaviours by addressing underlying sensory processing difficulties. These findings highlight the dual role of sensoriality in both exacerbating and mitigating socio-communicative challenges, underscoring the importance of tailored interventions that address sensory needs while promoting adaptive behaviour.

**Classroom Context**: The classroom context is a significant environment where sensory processing behaviours and challenges can manifest and impact children’s educational experiences. Several studies in this cluster have focused on assessing sensory processing difficulties in the classroom setting ([28]; [66]). Researchers have compared the sensory profiles of children with developmental disorders, such as ASD and ADHD, with typically developing children in the classroom context ([28]; [66]). These investigations have revealed that children with ASD and ADHD often exhibit more pronounced sensory processing challenges in the classroom environment compared to home settings ([28]; [66]).

Additionally, some studies have explored the effectiveness of classroom-based interventions and strategies for supporting children with sensory processing difficulties ([11]; [18]). For example, Bagatell et al. examined the use of therapy ball chairs and their impact on classroom participation for children with ASD ([11]). The findings highlighted the importance of considering each child’s unique sensory processing patterns and using sound clinical reasoning when recommending sensory strategies in the classroom ([11]). Furthermore, researchers have investigated the relationship between sensory processing challenges and academic performance or participation in the classroom setting ([63]). This study contributes to our understanding of how sensory processing difficulties can influence children’s educational experiences and highlight the need for appropriate support and accommodations in the classroom context ([63]).

**Using EEG Technology**: This cluster focuses on the use of electroencephalography (EEG) technology in the study of SPDs. Several studies have utilized EEG to investigate brain processing mechanisms in children with SPDs ([20]; [23]). Davies and Gavin employed EEG measures to examine sensory gating, a process related to sensory integration, in children with SPDs and typically developing children ([23]). Their findings suggested that children with SPDs exhibited less sensory gating compared to their typically developing peers, providing empirical evidence for unique brain processing mechanisms in children with SPDs ([23]). Similarly, Chang et al. used EEG to assess autonomic and behavioural responses to auditory stimuli in children with ASD and typically developing children. The study revealed that children with ASD had higher resting skin conductance and stronger skin conductance reactivity to tones, indicating heightened sympathetic activation and potentially contributing to the understanding of auditory processing difficulties in ASD ([20]). These studies demonstrate the potential of EEG technology in validating the diagnosis of SPDs and understanding the underlying neurophysiological mechanisms associated with sensory processing difficulties ([20]; [23]). By providing objective measures of brain activity and sensory processing, EEG can complement behavioural observations and parent-reported measures, contributing to a more comprehensive assessment and understanding of SPDs ([20]; [23]).

**Sensory Processing Correlate**: This cluster explores the sensory processing correlates and their associations with various domains, including occupational performance, behaviour, and cognitive abilities. Several studies have investigated the relationship between sensory processing difficulties and occupational performance in children with developmental disorders, such as fragile X syndrome (FXS) and ASD ([13]). This study found significant correlations between sensory processing vulnerabilities and individual differences in occupational performance, including school participation, self-care, and play ([13]). Additionally, researchers have examined the associations between sensory processing and cognitive functions, such as attention and executive functioning ([28]). The authors found that sensory processing measures were significant predictors of ASD severity and ADHD symptoms in children with ASD ([28]). Furthermore, some studies have explored the sensory processing correlates of specific behaviours, such as challenging behaviour and stereotypy ([25]; [52]). Devlin et al. compared the effects of sensory integration therapy and behavioural interventions on rates of challenging behaviour in children with ASD, providing insights into the potential role of sensory processing in these behaviours ([25]).

By investigating the associations between sensory processing and various domains, these studies contribute to our understanding of the impact of sensory processing difficulties on children’s functioning and behaviour, informing the development of appropriate interventions and support strategies ([13]; [25]; [28]; [52]).

**Behavioural Responses**: This cluster explores behavioural responses related to sensory processing in various populations, including comparative studies between different groups. Several studies have examined behavioural responses to sensory stimuli in individuals with developmental disabilities, such as ASD and intellectual disabilities ([13]). Baranek et al. investigated the prevalence of sensory defensiveness, a type of atypical sensory response, in adults and children with developmental disabilities ([13]). The findings suggested that sensory defensiveness behaviours are prevalent in this population and may be related to a broader construct of sensory defensiveness. Additionally, researchers have investigated the relationship between behavioural responses and sensory processing patterns or subtypes ([47], [46]; [58]). Mailloux et al. examined patterns of sensory integrative dysfunction in children and found associations between specific patterns, such as tactile defensiveness and attention, and behavioural responses ([47]). Purpura et al. explored behavioural differences in sensory processing profiles between children with SPDs and those with ASD ([58]). By studying behavioural responses to sensory stimuli and their relationships with sensory processing patterns, these studies contribute to a better understanding of the manifestations and subtypes of sensory processing difficulties, which can inform assessment and intervention approaches ([13]; [47], [46]; [58]).

**Sensory Defensiveness**: Sensory defensiveness, also known as over-responsivity to sensory stimuli, is a key aspect of SPDs and has been explored in several studies within this cluster. Researchers have investigated the nature and prevalence of sensory defensiveness across different populations, including individuals with developmental disabilities ([13]) and children with ASD ([47]). Baranek et al. found that sensory defensiveness behaviours, such as tactile defensiveness and noise sensitivity, were prevalent in adults and children with developmental disabilities, with estimates ranging from 3% to 30% ([13]).

Additionally, studies have explored the relationship between sensory defensiveness and other aspects of functioning, such as occupational performance and adaptive behaviour ([15]; [16]). Bellefeuille et al. presented a case report demonstrating how occupational therapy using a sensory integrative approach addressed sensory over-responsivity affecting a child’s ability to acquire age-appropriate toileting habits ([15]). Berardi et al. investigated the psychometric properties of the Italian version of the Toileting Habit Profile Questionnaire—Revised in children with ASD, highlighting the potential association between sensory processing difficulties and defecation disorders ([16]).

Furthermore, researchers have examined the assessment and measurement of sensory defensiveness using various tools and approaches ([22]; [26]). Cohn and Cermak discussed the importance of including the family perspective in sensory integration outcomes research, emphasizing the need to examine how interventions influence child and family function at multiple levels ([22]). Dugas et al. compared the Sensory Profile and the Sensory Processing Measure–Home Form in assessing sensory features in children with ASD, contributing to the understanding of the convergent validity of these measures ([26]).

These studies shed light on the prevalence, manifestations, and assessment of sensory defensiveness, as well as its potential impact on occupational performance and adaptive behaviour, informing the development of appropriate interventions and support strategies ([13]; [15]; [16]; [22]; [26]; [47]).

**Sound-Based Intervention**: This cluster focuses on the use of sound-based interventions, particularly in the context of sensory processing and occupational therapy for children with developmental disorders. Several studies have explored the effectiveness of sound-based interventions in addressing sensory processing difficulties and related challenges ([31]). The authors conducted a single-subject case–control study investigating the effects of The Listening Program, a sound-based intervention, on a child with ASD and auditory sensory over-responsivity. The results indicated a decrease in negative and self-stimulatory behaviours after the intervention, suggesting potential benefits of sound-based approaches in reducing sensory-related behaviours ([31]).

Comparative studies have also been conducted to evaluate the efficacy of sound-based interventions in comparison to other interventions or approaches ([25]; [34]). Devlin et al. compared the effects of sensory integration therapy and a behavioural intervention on rates of challenging behaviour in children with ASD. The findings demonstrated that the behavioural intervention was more effective than sensory integration therapy in reducing challenging behaviour ([25]). Granados and Agís reviewed the theoretical bases of hippotherapy (therapy involving horses) and suggested that it can lead to psychological, social, and educational benefits for children with special needs by affecting multiple sensory systems, including the auditory system ([34]). Additionally, researchers have explored the potential mechanisms and theoretical frameworks underlying sound-based interventions ([45]). Ma and Lee investigated the use of fabric samples containing tactile and visual stimuli in combination with balls during parent–child interactions for children with ASD, drawing from SIT ([45]). These studies contribute to the understanding of sound-based interventions and their potential applications in addressing sensory processing challenges, challenging behaviours, and occupational performance in children with developmental disorders ([25]; [31]; [34]; [45]).

**SenITA-RCT**: This cluster focuses on the Sensory Integration Therapy for Children with ASD and Sensory Processing Difficulties (SenITA) randomized controlled trial (RCT). The SenITA RCT, conducted by Randell et al., aimed to determine the clinical effectiveness and cost-effectiveness of sensory integration therapy for children with ASD and sensory processing difficulties across behavioural, functional, and quality-of-life outcomes ([62]). The study employed a parallel-group RCT design, incorporating an internal pilot and a process evaluation, involving children with ASD from Wales and England. The findings of the SenITA RCT revealed no significant main effects of the sensory integration therapy intervention on the primary outcome measures, including problem behaviours and Clinical Global Impression Scale—Improvement scores, at 6 or 12 months post-randomization ([62]). However, subgroup differences were observed for irritability/agitation at 6 months based on the child’s sex and ADHD status ([62]).

Additionally, the study reported improvements in carer-rated goal performance and satisfaction across therapy sessions, suggesting potential benefits of sensory integration therapy for individualized performance goals ([62]). The health economic evaluation indicated that sensory integration therapy is unlikely to be cost-effective compared to usual care alone ([62]). While the SenITA RCT did not demonstrate clear clinical benefits of sensory integration therapy above standard care for the overall sample, the findings highlight the importance of considering individual differences and subgroups in response to interventions ([62]). The study contributes to the ongoing research efforts in understanding the effectiveness of sensory integration therapy for children with ASD and sensory processing difficulties.

The use and applicability of SIT are supported by international guidelines that emphasize evidence-based practices. For example, the American Occupational Therapy Association (AOTA) recommends SIT as part of a comprehensive intervention plan for children with sensory processing difficulties, particularly in the context of ASD and ADHD ([55]). Similarly, the World Federation of Occupational Therapists (WFOT) highlights the importance of individualized sensory interventions that address specific sensory needs while promoting functional outcomes.

These guidelines underscore the need for rigorous assessment and tailored interventions that consider the unique sensory profiles of each child. For instance, the (EASI) tool has been validated for assessing sensory perception and integration, providing a standardized framework for clinical practice ([46]). By adhering to these guidelines, clinicians can ensure that SIT is implemented in a manner that maximizes its effectiveness and aligns with best practice in occupational therapy.

**Rubinstein–Taybi syndrome (RTS)**: This cluster explores the application of SIT and interventions in individuals with RTS, a rare genetic disorder characterized by intellectual disability and developmental delays. A study by Balikci et al. reported a case of a 3-year-old child with RTS who received SIT intervention ([12]). The researchers evaluated the effects of SIT on the child’s sensory processing, motor functions, and parental goals. The findings indicated significant improvements in parental reports of sensory processing, particularly in areas such as vestibular, tactile, and oral functioning, as well as gains in functional motor skills and goal attainment ([12]). This contributes to the understanding of sensory processing challenges and the potential benefits of sensory integration interventions in individuals with RTS and other developmental disabilities. However, it is important to note that the research in this area is limited, and further investigations are needed to establish the efficacy and applicability of sensory integration approaches for individuals with RTS across different age groups and severity levels.

**Treatment**: This cluster focuses on the application and effectiveness of various treatments and interventions, particularly in the context of sensory processing and occupational therapy for children with developmental disorders. Several studies have investigated the efficacy of sensory integration therapy as a treatment approach for children with ASD and related conditions ([67]; [68]). Schaaf and Nightlinger presented a case report demonstrating the use of occupational therapy using a sensory integrative approach (OT-SI) and its impact on a child’s occupational performance and behaviour ([68]). The findings supported the theoretical underpinnings of SIT, suggesting that improvements in sensory processing abilities can influence adaptive behaviour and occupational performance.

Additionally, researchers have compared the effectiveness of sensory integration therapy with other interventions, such as behavioural approaches ([25]; [48]). Devlin et al. compared the effects of sensory integration therapy and a functional-based behavioural intervention on rates of challenging behaviour in children with ASD ([25]). The results indicated that the behavioural intervention was more effective in reducing challenging behaviour compared to sensory integration therapy. Furthermore, some studies have explored the potential benefits of incorporating sensory integration principles into alternative or complementary therapies, such as hippotherapy (therapy involving horses) ([34]). They reviewed the theoretical bases of hippotherapy and suggested that it can lead to psychological, social, and educational benefits for children with special needs by affecting multiple sensory systems ([34]). These studies contribute to the understanding of the effectiveness and potential applications of sensory integration therapy and related interventions in addressing sensory processing challenges, challenging behaviours, and occupational performance in children with developmental disorders.

**Paediatric Rehabilitation**: This small cluster appears to focus on the use of assessments and outcome measures in the context of paediatric rehabilitation, potentially including children with sensory processing difficulties or developmental disabilities. One study examined the feedback and perspectives of Malaysian parents on three proxy-rated assessments used in paediatric rehabilitation settings ([2]). These assessments likely aimed to evaluate various domains of functioning, such as sensory processing, behaviour, or adaptive skills, in children receiving rehabilitation services. While the specific details of the studies in this cluster are limited, the inclusion of such a cluster suggests research efforts towards evaluating and improving the assessment tools and processes used in paediatric rehabilitation settings, which may involve children with SPDs, ASD, or other developmental conditions ([2]). Obtaining feedback from parents or caregivers on the usability, cultural relevance, and appropriateness of assessment measures is crucial in ensuring the accurate and meaningful evaluation of children’s abilities and needs within rehabilitation contexts. Such studies can inform the refinement or development of assessments tailored to specific populations or cultural contexts, ultimately enhancing the quality of care and support provided to children in paediatric rehabilitation programs.

## 4. Discussion

The purpose of this study was to examine the SPM and SIT by employing both scientometric and narrative methodologies. The results of the present study reveal diverse research themes and temporal patterns in the field of sensory processing and integration. There are three key findings:

First, the scientometric analysis identified 11 major clusters representing different aspects of sensory processing and integration research. The largest cluster, ‘sensory processing behaviour’ (64 articles), focused on comparative studies examining sensory processing behaviours and challenges across various domains ([28]; [66]). Other prominent clusters included ‘Classroom Context’ (53 articles), exploring sensory processing difficulties in educational settings ([28]; [66]), and ‘Using EEG Technology’ (53 articles), utilizing electroencephalography to investigate brain processing mechanisms in children with sensory processing disorders ([23]).

Second, the narrative analysis provided qualitative insights into the studies within each cluster, highlighting the theoretical frameworks, methodological approaches, and contextual factors influencing the research. For instance, the ‘Sensory Defensiveness’ cluster (30 articles) explored the nature and prevalence of sensory defensiveness, its relationship with occupational performance and adaptive behaviour, and the assessment and measurement of sensory defensiveness ([13]; [15]; [16]; [22]; [26]; [47]). Third, the study identified emerging research areas, such as the application of sensory integration theory in mental health contexts ([77]) and the use of technology in designing sensory-friendly environments ([79]).

This pattern of results is consistent with the previous literature, which has highlighted the diverse applications of the SPM and SIT across various populations, settings, and research domains ([42]; [53]). These results are consistent with the claim that sensory integration theory provides a valuable framework for understanding and addressing sensory processing challenges ([1]; [39]). Whereas past researchers have found that the SPM has been primarily used in paediatric populations ([41]; [49]), the present study has shown its applicability across the lifespan, including adolescent and adult populations ([69]).

Our findings highlight the value of combining scientometric and narrative synthesis methodologies in understanding the research of sensory processing and integration. The scientometric analysis provided a quantitative overview of publication trends, and thematic developments, while the narrative synthesis offered qualitative insights into the theoretical underpinnings, methodological approaches, and contextual factors influencing the studies. It is interesting that the cluster ‘SenITA-RCT’ (24 articles) focused on the Sensory Integration Therapy for Children with ASD and Sensory Processing Difficulties randomized controlled trial ([62]), which aimed to determine the clinical effectiveness and cost-effectiveness of sensory integration therapy. This finding may be explained by the idea that there is an increasing emphasis on evidence-based practice and the need for rigorous evaluation of interventions in the field of sensory processing and integration. Taken together, our findings indicate growing interest and research activity in the field of sensory processing and integration, with a focus on advancing theoretical frameworks, developing assessment tools, and evaluating interventions across diverse populations and settings.

While many studies support the effectiveness of SIT and the SPM, it is important to acknowledge the existing criticism and mixed results in the literature. For example, some researchers have questioned the methodological rigour of SIT studies, noting issues such as small sample sizes, a lack of control groups, and variability in intervention protocols ([14]; [17]). Additionally, the effectiveness of SIT has been challenged by studies that found no significant differences between SIT and other interventions ([25]; [62]). These findings highlight the need for more rigorous research to clarify the role of SIT in addressing sensory processing difficulties and to explore alternative approaches that may complement or enhance its effectiveness.

The identification of clusters such as ‘Classroom Context’ and ‘Using EEG Technology’ reflects emerging trends in sensory processing research. However, it is important to consider the strength and reliability of these clusters. For example, the ‘Classroom Context’ cluster (silhouette score = 0.913) demonstrates high internal consistency, suggesting a strong thematic focus on sensory processing in educational settings. Similarly, the ‘Using EEG Technology’ cluster (silhouette score = 0.842) highlights the growing use of neurophysiological tools to study sensory processing disorders. While these clusters provide valuable insights, further empirical research is needed to validate their findings and explore their implications for clinical practice.

While the narrative synthesis provides valuable qualitative insights, it is important to acknowledge potential biases in the selection process. The purposive sampling strategy, while necessary to ensure relevance and depth, may introduce selection bias, as it prioritizes studies that align with the research questions. To mitigate this, we employed clear and transparent selection criteria, including methodological rigour and availability of full texts, and included studies with diverse perspectives and research designs. Additionally, the narrative synthesis was guided by the clusters identified in the scientometric analysis, ensuring a systematic and data-driven approach to study selection.

### 4.1. Limitations

There are at least three potential limitations concerning the results of this study. The first limitation concerns the search strategy and database selection. While we aimed to be comprehensive in our search, it is possible that relevant studies may have been inadvertently excluded due to the specific search terms or databases used. The second potential limitation is that the narrative synthesis was guided by the clusters generated from the scientometric analysis, which could have introduced biases or limitations inherent to the clustering algorithms and methodologies employed. Although the present research cannot rule out these explanations, it seems useful to point out issues that may conflict with these results. Besides our own interpretation of the data, an additional explanation warrants comment. For example, researchers have suggested that the narrative synthesis process may have been influenced by the availability and depth of information provided in the included studies ([1]; [33]; [53]; [71]).

### 4.2. Implications

Despite these limitations, these results suggest several theoretical and practical implications. From a theoretical perspective, the findings contribute to a growing body of evidence suggesting that sensory integration theory provides a comprehensive framework for understanding and addressing sensory processing challenges across various domains and populations ([1]; [53]). These data also have some potential intervention implications. For example, the identification of emerging research areas, such as the application of sensory integration theory in mental health contexts ([77]) and the use of technology in designing sensory-friendly environments ([79]), highlights the potential for expanding the scope of sensory integration interventions and support strategies beyond traditional domains.

### 4.3. Future Directions

The present study represents an initiative attempt to address these issues by combining scientometric and narrative methodologies. We feel that further research examining the long-term impacts of sensory processing difficulties and interventions through longitudinal studies may shed light on the developmental trajectories of sensory processing challenges and the sustained effects of interventions. Although these studies support the value of interdisciplinary collaborations in advancing the understanding and practice of sensory integration ([42]; [61]), their most important contribution may be that they raise a variety of intriguing questions for future study, such as the comparative effectiveness of different interventions and the cultural adaptations and validations of assessment tools and interventions for diverse populations.

In terms of future research, it would be useful to extend the current findings by examining the potential of qualitative and mixed-methods studies to capture the lived experiences, perspectives, and subjective aspects of sensory processing and integration. If, as the present study suggests, sensory integration theory provides a valuable framework for understanding and addressing sensory processing challenges, then there is a need for research that explores the integration of sensory integration principles into educational environments and the development of evidence-based practices and accommodations to support children with sensory processing challenges in their academic pursuits.

## 5. Conclusions

The present research, therefore, contributes to a growing body of evidence suggesting that the sensory processing measure and sensory integration theory offer diverse applications across various populations, settings, and research domains. We hope that the current research will stimulate further investigation of this key area, fostering interdisciplinary collaborations and promoting the development of innovative interventions and support strategies for individuals with sensory processing challenges.

## Figures and Tables

**Figure 1 behavsci-15-00395-f001:**
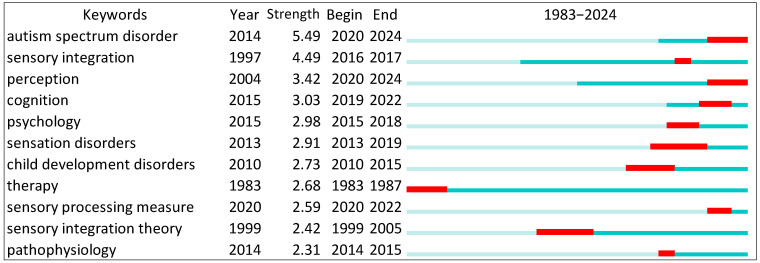
Top 11 keywords with the strongest citation bursts.

**Figure 2 behavsci-15-00395-f002:**
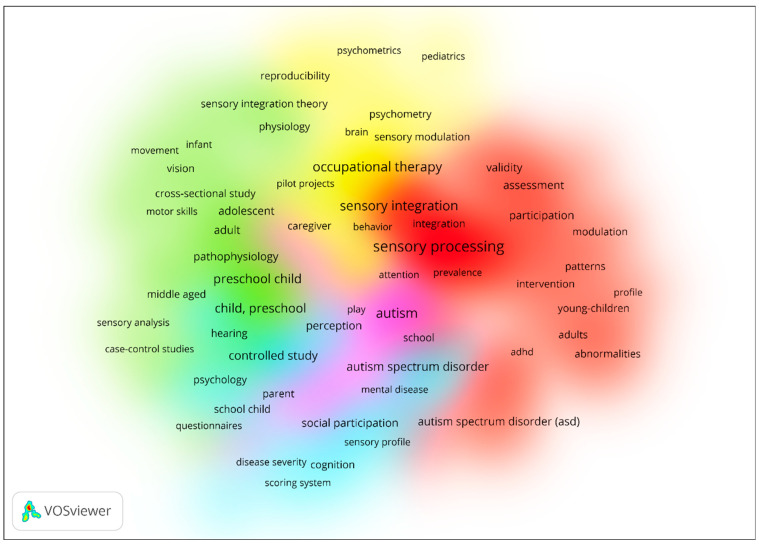
Density visualization for keyword co-occurrences in the sensory processing measure and sensory integration theory.

**Table 1 behavsci-15-00395-t001:** Search terms 26 November 2024.

Query	Database	Search Terms	Results
1	Scopus	(TITLE(“ayres sensory integration theory”) OR TITLE(“sensory integration theory”)) OR (TITLE(“sensory processing measure”)) OR (TITLE-ABS-KEY(“sensory processing measure”)) OR (TITLE-ABS-KEY(“ayres sensory integration theory”) OR TITLE-ABS-KEY(“sensory integration theory”))	146
2	Web of Science Core Collection	TI = (“sensory processing measure”)	11
3		TS = (“sensory processing measure”)	57
4		(TI = (“ayres sensory integration theory”)) OR TI = (“sensory integration theory”)	11
5		(TS = (“ayres sensory integration theory”)) OR TS = (“sensory integration theory”)	38
6		#1 OR #2 OR #3 OR #4	95
1 and 6	Both	Merged from the two databases	241
After duplicates removal			238
Included in the analysis (considered disqualified in CiteSpace)			215
Included in VOSviewer			238

**Table 2 behavsci-15-00395-t002:** Summary of the largest clusters.

Cluster	Size	Silhouette	Label (LSI)	Label (LLR)	Label (MI)	Average Year
Sensory Processing Behaviour	64	0.691	comparative study	sensory processing behaviour	sensory processing challenge (0.34)	2010
Classroom Context	53	0.913	safety cue	classroom context	open space (1.32)	2018
Using EEG Technology	53	0.842	using structural equation modelling	using EEG technology	open space (0.07)	2012
Sensory Processing Correlate	51	0.775	sex difference	sensory processing correlate	open space (0.66)	2004
Behavioural Responses	31	0.798	comparative study	behavioural responses	open space (0.33)	2009
Sensory Defensiveness	30	0.889	toileting habit profile	sensory defensiveness	open space (0.8)	1998
Sound-Based Intervention	29	0.843	comparative study	sound-based intervention	open space (0.54)	2015
SenITA-RCT	24	0.934	e-waste exposure	SenITA-RCT	ASD (0.06)	2020
Rubinstein–Taybi syndrome (RTS)	18	0.931	school-aged children	Rubinstein–Taybi	ASD (0.05)	2001
Treatment	10	0.966	effect	treatment	ASD (0.07)	2007
Paediatric Rehabilitation	5	0.984	Malaysian parents’ feedback on three proxy-rated assessments used in paediatric rehabilitation	paediatric rehabilitation	ASD (0.08)	2016

## Data Availability

The data presented in this study are available on request from the corresponding authors.

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
