# Peer review of "Sensory Processing Measure and Sensory Integration Theory: A Scientometric and Narrative Synthesis"

_behavsci, 2025, doi:10.3390/bs15030395_

Round 1

Reviewer 1 Report

Comments and Suggestions for Authors

Dear Authors,

Just a few comments, for a very well done work, full of ideas, useful for an updated critical reflection on the literature and for clinical practice.
1) Better argue the relationship between socio-communicative behavior and sensoriality: as regards ASD and ADHD, sensory overload negatively impacts both stress (in terms of autonomic reactions, there is some literature on this; non univocal results, but there are) and the way in which we adapt our behavior in response to the variation of environmental stimuli; on the other hand, the search for stimuli or the self-regulation of stimulation, whether done in a naive or conscious way, can help reduce overload or increase emotional tone, improving sociability. Therefore, describe these points better and enrich the bibliography.
2) Insert, if available, indications of international guidelines on the use and applicability of SIT;
3) Perhaps, for the narrative collection, a selection criterion could be added to expand what has been collected on the basis of the "title" criterion.

Author Response

Dear Authors,

Just a few comments, for a very well done work, full of ideas, useful for an updated critical reflection on the literature and for clinical practice.

Dear Colleague,

Thank you for your thoughtful and constructive feedback on our manuscript. We appreciate your positive remarks and have carefully addressed your suggestions to enhance the quality and depth of our work. Below, we outline the changes made in response to your comments. All our changes are in red in the revised version. 

1) Better argue the relationship between socio-communicative behavior and sensoriality: as regards ASD and ADHD, sensory overload negatively impacts both stress (in terms of autonomic reactions, there is some literature on this; non univocal results, but there are) and the way in which we adapt our behavior in response to the variation of environmental stimuli; on the other hand, the search for stimuli or the self-regulation of stimulation, whether done in a naive or conscious way, can help reduce overload or increase emotional tone, improving sociability. Therefore, describe these points better and enrich the bibliography.

We have expanded the discussion on the relationship between socio-communicative behavior and sensoriality, particularly in the context of ASD and ADHD.

2) Insert, if available, indications of international guidelines on the use and applicability of SIT;

We have included a section on international guidelines for the use and applicability of SIT. 

3) Perhaps, for the narrative collection, a selection criterion could be added to expand what has been collected on the basis of the "title" criterion.

We have refined the selection criteria for the narrative synthesis to ensure a more systematic and comprehensive review of the literature. 

Thank you so much for all your comments and support to improve our manuscript. We remain committed to any further suggestions towards the best of this manuscript.

Warm regards, 

Authors 

Reviewer 2 Report

Comments and Suggestions for Authors

The authors assumed a review of the literature on sensory integration using two methods of collecting and analyzing the acquired data (as complementary to each other). The article concerns a topic that has been discussed many times in the literature. After reading it, I have a few comments for the authors.
1. After reading the article, it is not clear what its purpose is: a comparison of two types of data analyses, a literature review or a comparison of two diagnostic tools. Neither in the introduction nor in the further structure of the article is it clearly exposed (and you can even notice that the authors are undecided on this issue)
2. There is no broader description of sensory integration disorders (causes from an interdisciplinary perspective, development of the dynamics of the disorder in relation to ASD, ADHD or other developmental difficulties cited by the authors).
3. References should be added to the statement "Research has used this measure to examine sensory differences 225 in specific conditions such as ASD, ADHD, FASD, and DCD, as well as in typically developing children."
4. Diagnostic tools are not described in detail.
5. It was not explained why only two databases were used and e.g. PubMed (medical database), which would complement the topic of physiological aspects of sensory integration, was not taken into account?
6. It would be worth providing data such as the number of publications in each cluster. 
7. Figure 2 is not legible. It should be supplemented with numerical data. Otherwise it is a narrative analysis.

Comments on the Quality of English Language

no comments

Author Response

The authors assumed a review of the literature on sensory integration using two methods of collecting and analyzing the acquired data (as complementary to each other). The article concerns a topic that has been discussed many times in the literature. After reading it, I have a few comments for the authors.

Dear Colleague,

Thank you for your detailed and constructive feedback on our manuscript. We appreciate your insights and have carefully addressed each of your comments to improve the clarity, depth, and rigor of our work. Below, we outline the changes made in response to your suggestions.

  1. After reading the article, it is not clear what its purpose is: a comparison of two types of data analyses, a literature review or a comparison of two diagnostic tools. Neither in the introduction nor in the further structure of the article is it clearly exposed (and you can even notice that the authors are undecided on this issue)

We appreciate this observation and have revised the Introduction to explicitly state the purpose of the study. The purpose is to examine the Sensory Processing Measure and Sensory Integration Theory using both scientometric and narrative methodologies. This dual approach allows us to provide a comprehensive understanding of the current state of research in sensory processing and integration, identify key themes and emerging trends, and inform future research directions and clinical applications.

  1. There is no broader description of sensory integration disorders (causes from an interdisciplinary perspective, development of the dynamics of the disorder in relation to ASD, ADHD or other developmental difficulties cited by the authors).

We have added a detailed description of sensory integration disorders, including their causes, neurological underpinnings, and developmental dynamics in relation to ASD, ADHD, and other conditions. Please kindly note that we cannot the paper longer than this since it is already over 12 thousand words.

  1. References should be added to the statement "Research has used this measure to examine sensory differences 225 in specific conditions such as ASD, ADHD, FASD, and DCD, as well as in typically developing children."

Please note this paragraph is just a summary for the introduction sections. We added a clarification.

  1. Diagnostic tools are not described in detail.

We have added a detailed description of the SPM and EASI, including their purpose, structure, and clinical applications.

  1. It was not explained why only two databases were used and e.g. PubMed (medical database), which would complement the topic of physiological aspects of sensory integration, was not taken into account?

We have added an explanation for the choice of databases, noting that Scopus and Web of Science were selected for their interdisciplinary coverage and the fact that almost all PubMed publications are included in these databases. We also clarify that our focus was broader than medical-specific literature.

  1. It would be worth providing data such as the number of publications in each cluster. 

Thank you. Please check Table 2, second column, size of each cluster. We coloured this in red in our revised version.

  1. Figure 2 is not legible. It should be supplemented with numerical data. Otherwise it is a narrative analysis.

We are sorry regarding Figure 2. We checked our provided version and it is the highest resolution available in the software. It appears that its quality decreased when was copied by the production team. We will make sure with them to have the original version kept. Thank you so much for withdrawing our attention to this.

Comments on the Quality of English Language

no comments

Thank you for your feedback on the quality of the English language in our manuscript. We noticed that while you selected the option “The English could be improved to more clearly express the research” in the review platform, you also mentioned in your report that you had “no comments” regarding the quality of the English language. We acknowledge that there could have been a chance you intended to select the alternative option, “The English is fine and does not require any improvement,” given your explicit statement in the report.

However, in the spirit of ensuring the highest standards of clarity and readability, we have carefully reviewed the entire manuscript. We have refined some  sections to improve the flow, grammar, and coherence of the text, ensuring that the research is communicated effectively to an international readership. We hope these revisions address any potential concerns and enhance the overall quality of the manuscript.

Thank you again for your valuable feedback and for bringing this to our attention.

We remain committed to make the best of this manuscript and to follow your guidance towards contributing to the scientific community with high quality science production. 

Authors

Round 2

Reviewer 2 Report

Comments and Suggestions for Authors

Dear Authors,
thank you for the corrections and additions made. I appreciate it and they are fully satisfactory.
Regarding the language of the publication, I actually wanted to point out that it does not require correction. However, thank you for checking the article again from this angle.
I have no further comments.

Author Response

Dear Colleague, 

Thank you so much for all your support throughout this process. 

Warm regards, 

Authors